# Quantitative proteomics analysis reveals possible anticancer mechanisms of 5'-deoxy-5'-methylthioadenosine in cholangiocarcinoma cells

Kankamol Kerdkumthong[1☯], Sutthipong Nanarong[1,2☯], Sittiruk Roytrakul[3], Thanawat Pitakpornpreecha[1,4], Phonprapavee Tantimetta[1], Phanthipha Runsaeng[1,4], Sumalee Obchoei[1,4]*

1 Faculty of Science, Division of Health and Applied Sciences, Biochemistry Graduate Program, Prince of Songkla University, Hatyai, Songkhla, Thailand, 2 Faculty of Pharmaceutical Sciences, Department of Pharmacognosy and Pharmaceutical Botany, Prince of Songkla University, Hatyai, Songkhla, Thailand, 3 National Center for Genetic Engineering and Biotechnology, National Science and Technology Development Agency, Pathumtani, Thailand, 4 Faculty of Science, Center of Excellence for Biochemistry, Prince of Songkla University, Hatyai, Songkhla, Thailand

☯ These authors contributed equally to this work.
* sumalee.o@psu.ac.th

## Abstract

Cholangiocarcinoma (CCA) is an aggressive cancer originating from bile duct epithelium, particularly prevalent in Asian countries with liver fluke infections. Current chemotherapy for CCA often fails due to drug resistance, necessitating novel anticancer agents. This study investigates the potential of 5'-deoxy-5'-methylthioadenosine (MTA), a naturally occurring nucleoside, against CCA. While MTA has shown promise against various cancers, its effects on CCA remain unexplored. We evaluated MTA's anticancer activity in CCA cell lines and drug-resistant sub-lines, assessing cell viability, migration, invasion, and apoptosis. The potential anticancer mechanisms of MTA were explored through proteomic analysis using LC-MS/MS and bioinformatic analysis. The results show a dose-dependent reduction in CCA cell viability, with enhanced effects on cancer cells compared to normal cells. Moreover, MTA inhibits growth, induces apoptosis, and suppresses cell migration and invasion. Additionally, MTA enhanced the anticancer effects of gemcitabine on drug-resistant CCA cells. Proteomics revealed the down-regulation of multiple proteins by MTA, affecting various molecular functions, biological processes, and cellular components. Network analysis highlighted MTA's role in inhibiting proteins related to mitochondrial function and energy derivation, crucial for cell growth and survival. Additionally, MTA suppressed proteins involved in cell morphology and cytoskeleton organization, important for cancer cell motility and metastasis. Six candidate genes, including ZNF860, KLC1, GRAMD1C, MAMSTR, TANC1, and TTC13, were selected from the top 10 most down-regulated proteins identified in the proteomics results and were subsequently verified through RT-qPCR. Further, KLC1 protein suppression by MTA treatment was confirmed through Western blotting. Additionally, based on TCGA data, KLC1 mRNA was found to be upregulated in the tissue of CCA patients compared to that of normal adjacent tissues. In summary, MTA shows promising anticancer

**Data Availability Statement:** All relevant data are within the paper and its Supporting information files. The MS proteomics data from this research have been deposited in the ProteomeXchange Consortium (http://www.proteomexchange.org/) through the jPOSTrepo partner (https://repository.jpostdb.org/entry/JPST002497), with dataset identifier PXD049119.

**Funding:** This research was supported by the National Science, Research and Innovation Fund (NSRF) and Prince of Songkla University (Ref. No. SCI6701024S) to Sumalee Obchoei. Kankamol Kerdkumthong was supported by a Prince of Songkla University-Ph.D. Scholarship (PSU_PHD2561-001). Phonprapavee Tantimetta was supported by a Prince of Songkla University-Ph.D. Scholarship (PSU_PHD2565-002). The funders had no role in the study design, data collection and analysis, decision to publish, or preparation of the manuscript.

**Competing interests:** The authors have declared that no competing interests exist.

potential against CCA by inhibiting growth, inducing apoptosis, and suppressing migration and invasion, while enhancing gemcitabine's effects. Proteomic analysis elucidates possible molecular mechanisms underlying MTA's anticancer activity, laying the groundwork for future research and development of MTA as a treatment for advanced CCA.

## Introduction

Cholangiocarcinoma (CCA) is an aggressive cancer that originates from the bile duct epithelium [1]. While it is considered rare globally, it is more prevalent in certain Asian countries, with the highest incidence in Thailand, where there is a high prevalence of the causative liver fluke infection [2]. Over the past few decades, both the incidence and mortality rates of CCA have significantly increased worldwide [3–5]. CCA tends to develop silently but exhibits a high level of invasiveness [6]. Consequently, it often remains asymptomatic in its early stages and is diagnosed at an advanced stage. Patients with metastatic CCA are poor candidates for surgery, and as a result, the common treatment for these patients is chemotherapy. Unfortunately, CCA shows a high resistance to chemotherapeutic drugs, leading to ineffective treatments and poor overall survival rates [7]. Even when a combination of chemotherapeutic drugs is used, the survival of patients with CCA remains poor. Therefore, there is an urgent to pursue new anticancer agents that are effective with minimal side effects.

5'-deoxy-5'-methylthioadenosine (MTA) is a naturally occurring sulfur-containing nucleoside (Fig 1A). It is found in low concentrations in prokaryotes, yeasts, plants, and eukaryotes. In mammals, including humans, MTA is a by-product of polyamine biosynthesis and plays a crucial role in various metabolic pathways, particularly the salvage pathways of methionine and purine [8,9]. Numerous reports have highlighted the physiological and pharmacological

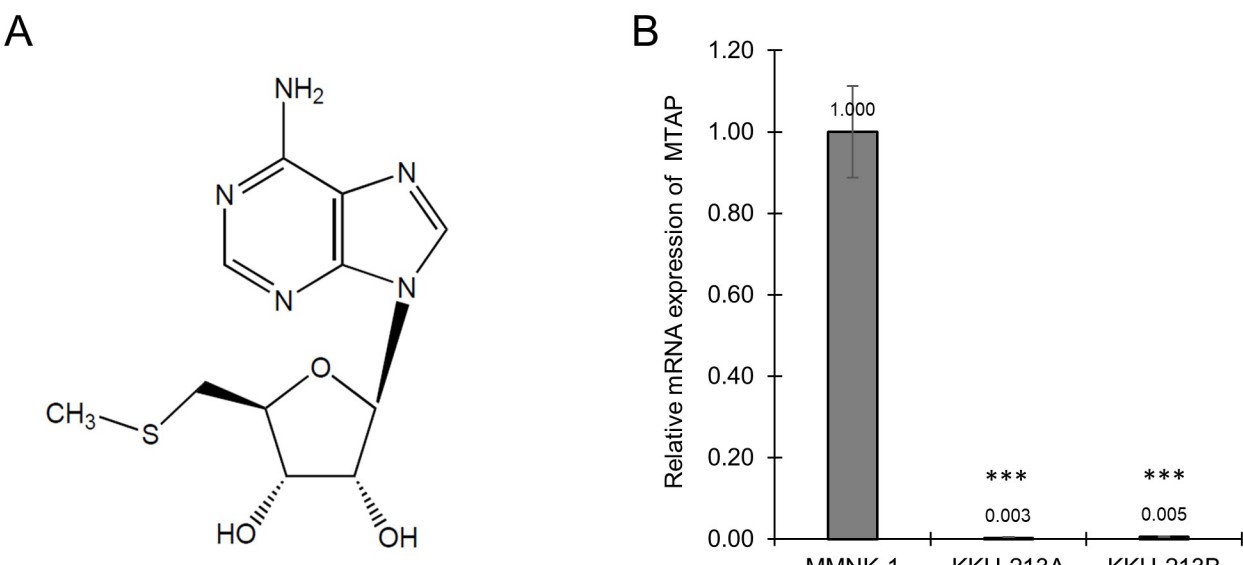

**Fig 1. Chemical structure of 5'-deoxy-5'-methylthioadenosine (MTA) and the expression of 5-methylthioadenosine phosphorylase (MTAP) in cell lines.** (A) Chemical structure of MTA. (B) relative MTAP mRNA expression in normal cell line (MMNK-1) and CCA cell lines (KKU-213A and KKU-213B).

roles of MTA, including anticancer activities, such as its ability to inhibit cell proliferation and invasion in multiple cancer types. For instance, studies have demonstrated that MTA can hinder the growth of cervical cancer, lymphoma, erythroleukemia, and promyelocytic cells [9,10]. Moreover, MTA has been shown to suppress the migration and invasion of colorectal cancer, liver cancer, and HT1080 fibrosarcoma cells [10,11]. Numerous studies have demonstrated that MTA's impact on cancer cells is notably more pronounced than on normal cells. This disparity arises due to the absence of 5-methylthioadenosine phosphorylase (MTAP), a crucial enzyme in the methionine salvage pathway, within cancer cells [10,12–14]. This deficiency disrupts the adenine salvage pathway, leading to the buildup of unmetabolized MTA and subsequent cellular toxicity. Despite extensive research on MTA's targets and mechanisms of action in various cancers, its effects on CCA remain unexplored. In this study, we investigated the impact of MTA on CCA cell lines, including those resistant to chemotherapeutic drug, gemcitabine and evaluated its toxicity on normal bile duct cell line. Additionally, we studied the effect of combination treatment of gemcitabine and MTA on CCA cell lines. Finally, we explored the possible anticancer mechanisms of MTA through quantitative proteomics and bioinformatics analyses.

## Materials & methods

### Chemicals and reagents

5′-deoxy-5′-methylthioadenosine (MTA), gemcitabine, and 3-(4,5-dimethylthiazol-2-yl)-2,5-diphenyltetrazolium bromide (MTT) reagents were purchased from Sigma (Sigma—Aldrich, St. Louis, MO, USA). Matrigel™ was purchased from BD Bioscience (BD Bioscience, Bedford, MA, USA).

### Cell culture

Two human CCA cell lines, KKU-213A and KKU-213B, were established from tissues of Thai CCA patients [15]. An immortalized human cholangiocyte cell line (MMNK-1) was established as previously described [16]. These cell lines were obtained from the Japanese Collection of Research Bioresources (JCRB) Cell Bank (Osaka, Japan). The gemcitabine-resistant cells, KKU-213A-GR and KKU-213B-GR, were established from the CCA parental cell lines, KKU-213A and KKU-213B, respectively, as previously described [17]. All cells were cultured in Dulbecco's modified Eagle's medium (DMEM) supplemented with 10% fetal bovine serum (FBS) (Gibco, NY, USA) and 1% antibiotics/antimycotics (Capricorn Scientific), in a humidified atmosphere of 5% $CO_2$ at 37˚C.

### Cell viability assay

Cell viability was determined using the MTT assay. Briefly, $3 \times 10^3$ cells per well (for 72-hour treatment) and $1 \times 10^3$ cells per well (for 7-day treatment) were seeded in a 96-wells plate and incubated overnight. The following day, the cells were exposed to various concentrations of MTA. After the specified incubation time, cell viability was determined using MTT assay. The percentage of cell viability and the half-maximal inhibitory concentration ($IC_{50}$) were then calculated and compared. To determine the effect of MTA combined with gemcitabine, the parental and gemcitabine resistant cells, which had been grown overnight, were treated with 6.25 and 12.5 μg/mL of MTA as a single agent or in combination with 3 μM gemcitabine for 72 hours. Cell viability was assessed using the MTT assay, as described above.

## Cell growth assay

The effect of MTA on cell growth was measured using MTT assay. KKU-213A and KKU-213B cells, at a density of $1.5 \times 10^3$ cells per well, were seeded in a 96-well plate, treated with 6.25, 12.5, and 25 µg/mL of MTA, and subjected to MTT assays on day 0, 2, 4, 6, and 8. The cell growth rate was calculated as a fold change compared to day 0.

## Apoptotic nuclear staining assay

Apoptotic cells were visualized under a microscope following a double staining procedure using two fluorescence DNA staining dyes, Hoechst 33342 and propidium iodide. Hoechst 33342 can permeate both live and dead cell membranes to stain DNA, while propidium iodide can only penetrate the membranes of dead cells. Nuclei that appear condensed and were stained with both dyes were counted as apoptotic nuclei. KKU-213A and KKU-213B cells ($2 \times 10^5$ cells/well) were seeded in a 6-well plate, incubated overnight, and then treated with 0, 12.5, and 25 µg/mL of MTA for 72 hours. The culture medium was replaced with a solution of Hoechst 33342 (5 µg/mL) and propidium iodide (10 µg/mL) for 5 minutes, while protected from light. Apoptotic cells were observed and counted under a fluorescence microscope (cell-Sens Standard software, Olympus IX73 inverted microscope, EVIDENT, Japan).

## Scratch assay

CCA cells ($2 \times 10^5$ cells/well) were seeded into a 12-well plate and cultured in complete medium overnight. When the cell monolayer reached approximately 90–100% confluence, wounds were created using a sterile pipette tip. Subsequently, the old medium and cell debris were removed, and the cells were incubated with 0, 12.5, and 25 µg/mL of MTA for 48 hours. Wound images were captured at 12, 24, 36, and 48-hour time points. The percentage of wound closure was measured using ImageJ software (version 1.53e, National Institute of Health, Bethesda, MD, USA).

## Cell migration and invasion assay

Cell migration and invasion assays were conducted using a modified Boyden chamber method with Transwell® inserts featuring 8.0 µm pore size (Corning, NY, USA). KKU-213A and KKU-213B cells ($2 \times 10^4$ cells) were seeded into the upper compartment of the Transwell® inserts, either in serum-free medium with or without MTA (12.5 and 25 µg/mL). For the invasion assay, the Transwell® inserts were pre-coated with 0.5 mg/mL Matrigel™ before use. In the lower chamber, complete medium was added to serve as a chemoattractant for cell migration and invasion. After 12 hours of incubation, non-migrated/non-invaded cells in the upper chamber were removed. The migrated and invaded cells were fixed with 4% paraformaldehyde, stained with 0.4% sulforhodamine B (SRB), and images were captured using a microscope. The numbers of migrated and invaded cells from 6 random low-power fields were counted and compared.

## Sample preparation for proteomics analysis

MTA-treated KKU-213A cells and untreated control cells were harvested via trypsinization and washed twice with 1x phosphate buffered saline (PBS). The cell pellets were lysed with 0.5% SDS and then centrifuged at 10,000xg for 15 minutes. The resulting supernatant was transferred to a new tube, mixed with two volumes of cold acetone, and incubated overnight at -20 ºC. The mixture was then thawed and centrifuged at 10,000xg for 15 minutes. After discarding the supernatant, the pellet was dried and stored at -80˚C until use.

## Liquid chromatography and tandem mass spectrometry

To identify the proteins affected by MTA treatment, we conducted proteomic analysis using liquid chromatography with tandem mass spectrometry (LC-MS/MS). The protein concentration of all samples was determined by the Lowry assay using BSA as the standard protein. To reduce disulfide bonds, we added 10 mM dithiothreitol in 10 mM ammonium bicarbonate to the protein solution, and the reformation of disulfide bonds in the proteins was blocked by alkylation with 30 mM iodoacetamide in 10 mM ammonium bicarbonate. Protein samples were prepared and digested with sequencing-grade porcine trypsin (at a ratio of 1:20) for 16 hours at 37°C. Subsequently, they were injected into an Ultimate3000 Nano/Capillary LC System (Thermo Scientific, UK) coupled with an HCTUltra LC-MS system (Bruker Daltonics Ltd, Hamburg, Germany), equipped with a Nano-captive spray ion source. Briefly, 5 µL of peptide digests were enriched on a µ-Precolumn 300 µm I.D. x 5 mm C18 Pepmap 100, 5 µm, 100 Å (Thermo Scientific, UK), separated on a 75 µm I.D. x 15 cm and packed with Acclaim PepMap RSLC C18, 2 µm, 100 Å, nanoViper (Thermo Scientific, UK). The C18 column was enclosed in a thermostatted column oven set to 60°C. Solvent A and B containing 0.1% formic acid in water and 0.1% formic acid in 80% acetonitrile, respectively, were supplied on the analytical column. A gradient of 5–55% solvent B was used to elute the peptides at a constant flow rate of 0.30 µL/min for 30 minutes. Electrospray ionization was carried out at 1.6 kV using the CaptiveSpray. Nitrogen was used as a drying gas (flow rate about 50 L/h). Collision-induced-dissociation (CID) product ion mass spectra were obtained using nitrogen gas as the collision gas. Mass spectra (MS) and MS/MS spectra were obtained in the positive-ion mode at 2 Hz over the range (m/z) 150–2200. The collision energy was adjusted to 10 eV as a function of the m/z value. The LC-MS analysis of each sample (100 ng/mL) was done in triplicate. The MS proteomics data from this research have been deposited in the ProteomeXchange Consortium (http://www.proteomexchange.org/) through the jPOSTrepo partner (https://repository.jpostdb.org/entry/JPST002497), with dataset identifier PXD049119.

## Protein identification and quantification

The LC-MS data were processed using DecyderMS [18,19] and searched against all nonredundant protein sequences from the UniProt database, which includes entries for *Homo sapiens* proteins, comprising 20,386 entries from Swiss-Prot and 181,774 entries from TrEMBL. Proteins affected by MTA treatment, exhibiting a decrease or increase in expression by at least 2-fold, with a false discovery rate (FDR) of less than 0.05, were identified as differentially expressed proteins (DEPs). In this study, the down-regulated proteins were further analyzed. The heatmap displaying the down-regulated proteins in MTA-treated KKU-213A cells compared to the untreated control cells was generated using MultiExperiment Viewer (MeV) software, following the method described by Howe and colleagues [20]. It was based on the Log2 values of protein expression in KKU-213A cells with or without MTA treatment.

## Protein function enrichment and cluster analysis

The PANTHER classification system (version 18.0, http://www.pantherdb.org) was employed for protein identification and classification across molecular function, biological process, cellular component, protein class, and signaling pathway [21]. To accomplish this, protein IDs (UniProtKB) of the down-regulated proteins in MTA-treated KKU-213A cells were submitted to PANTHER, with Homo sapiens selected as the organism. Gene Ontology (GO) enrichment analysis on the down-regulated proteins was conducted using the ShinyGo web server [22], version 0.741, utilizing the Homo sapiens proteome as a reference for analysis. The down-regulated proteins were then categorized according to the GO biological process and GO cellular

**Table 1. Lists of gene specific primers for qPCR.**

| Gene | | Sequence (5′to3′) |
|---|---|---|
| ACTB | For: | GGATTCCTATGTGGGCGACG |
| | Rev: | TTGTAGAAGGTGTGGTGCCAG |
| MTAP | For: | AGCACACCATCATGCCTTCA |
| | Rev: | AATGACAATATCGCCGGGCT |
| ZNF860 | For: | ACACTCCCTCTCTCGGTCTTC |
| | Rev: | CCTCCCTGAGCTGTGACTTTG |
| KLC1 | For: | TGAGGTCTCGTAAACAGGGT |
| | Rev: | AAGATCAGTGCCATCTTCCTCC |
| GRAMD1C | For: | ACTGTCCGTCAGGTGATGAAT |
| | Rev: | AAAGCTCCAGTCACCACTCC |
| MAMSTR | For: | GTCTCCGATCCTGTGTCTGC |
| | Rev: | CTGCTGGAGGATCCATGGTG |
| TANC1 | For: | GTGTGTCTGCTGACCAAGAAGG |
| | Rev: | GACCACTCACAAGTCAGCAGGT |
| TTC13 | For: | CATGACTCAGACTGCGAACC |
| | Rev: | ATCAGTGGCAAACGGGAATC |

component collections. An FDR of less than 0.05 was considered significant. In the network analysis, each node represented an enriched GO term, with connected GO terms linked by lines. The thickness of each line indicated the percentage of overlapping genes, while the size of the node was proportional to the number of genes it represented.

## RT-qPCR

Reverse transcription quantitative polymerase chain reaction (RT-qPCR) was used to measure mRNA expression of genes of interest. After appropriate cell culture conditions or treatments, total RNA was extracted from cells using TRIzol™ reagent (Thermo Fisher Scientific, USA). Then, 2 μg total RNA of each sample was converted into cDNA using the RevertAid First Strand cDNA Synthesis Kit (Thermo Fisher Scientific, Vilnius, Lithuania). The obtained cDNA was then amplified using gene-specific primers (Table 1) and Maxima SYBR Green/ROX qPCR Master Mix (Thermo Fisher Scientific, Vilnius, Lithuania), following the manufacturer's protocol. The relative mRNA expression was normalized using the internal control gene (ACTB) and calculated using the $2^{-\Delta\Delta Ct}$ equation.

## Western blotting

Following treatment with MTA at 0, 25, and 50 μg/mL for 48 hours, total protein was extracted from KKU-213A and KKU-213B cells using RIPA lysis buffer (Visual Protein, Taiwan) containing a protease inhibitor cocktail (Merck KGaA, Germany). The protein concentration was determined using a bicinchoninic acid protein concentration assay kit (Bio Basic, USA). Then, 30 μg of protein from each sample was separated using SDS-PAGE and transferred to polyvinylidene fluoride (PVDF) membranes. The membrane was incubated at 4˚C overnight with primary antibodies against KLC1 (1:1,000 dilution, Abcam, Cambridge, UK) or β-actin (1:10,000 dilution, Sigma-Aldrich, Darmstadt, Germany), followed by incubation with HRP-conjugated secondary antibodies (1:10,000 dilution, Sigma-Aldrich, Darmstadt, Germany) for 1 hour. Immunoreactive signals were developed via ECL prime reagent (Thermo Fisher

Scientific, USA) and captured with an Alliance®Q9-ATOM Chemiluminescence Imager (Uvitec, Cambridge, UK).

### Analysis of KLC1 mRNA expression in CCA patients' tissue

The expression level of KLC1 in CCA patient tissues was analyzed using the online analysis platform The Gene Expression Profiling Interactive Analysis 2 (GEPIA2) (http://gepia2.cancer-pku.cn/#analysis) based on The Cancer Genome Atlas Program (TCGA) data to compare its mRNA expression between CCA tissues and normal adjacent tissues.

### Statistical analysis

The results are presented as the mean ± SD from at least three replicates. Statistical significance was determined using Student's t-test, with $p < 0.05$ considered statistically significant.

## Results

### MTAP deficiency in CCA cell lines

The RT-qPCR analysis was conducted to examine the mRNA levels of MTAP in CCA cell lines and normal cell line. The results indicated a significant downregulation of MTAP expression in both CCA cell lines (KKU-213A and KKU-213B) when compared to the normal cell line (MMNK-1) (Fig 1B).

### MTA attenuates growth of CCA cells and enhances the effect of gemcitabine *in vitro*

To assess the cytotoxic effect of MTA on CCA cell lines and a normal cell line, KKU-213A, KKU-213B, and MMNK-1 were exposed to various concentrations of MTA, and cell viability was determined using the MTT assay. The results indicated that MTA reduced cell viability in a dose-dependent manner both at 72 hours (Fig 2A) and 7 days (Fig 2B) after treatment. Furthermore, MTA at concentrations of 12.5, 25, 50, and 100 μg/mL had a less pronounced effect on the viability of normal cells compared to cancer cell lines (Fig 2A and 2B). The half-maximal inhibitory concentration ($IC_{50}$) values of MTA in both CCA cell lines and the normal cell line are provided in Table 2. Additionally, a cell growth assay was performed using relatively low concentrations of MTA. The results demonstrated that MTA at a concentration of 25 μg/mL significantly inhibited the growth of both KKU-213A (Fig 2C) and KKU-213B (Fig 2D) on day 4, 6, and 8 compared to the untreated control. Meanwhile, MTA at 6.25 and 12.5 μg/mL inhibited the growth of KKU-213A (Fig 2C) and KKU-213B (Fig 2D) on day 6 and 8. Moreover, the antiproliferative effect of MTA on the gemcitabine-resistant CCA cell lines, KKU-213A-GR and KKU-213B-GR, was investigated. The results revealed that the combination of gemcitabine and MTA at concentrations of 6.25 and 12.5 μg/mL significantly reduced cell viability in both KKU-213A-GR and KKU-213B-GR, as well as in their parental counterparts. Notably, treatment with MTA alone had a stronger effect on the gemcitabine-resistant CCA cell lines than on the parental cell lines (Fig 2E and 2F).

### MTA induces apoptosis of CCA cells

To determine whether MTA could induce apoptotic cell death, CCA cell lines were treated with 12.5 and 25 μg/mL of MTA for 72 hours and double-stained with Hoechst and propidium iodide. The cells were observed and photographed under a fluorescence microscope (Fig 3A and 3C). Apoptotic nuclei and total nuclei in each treatment group were counted from six randomly selected low-power fields, and the percentage of apoptotic nuclei was calculated. The

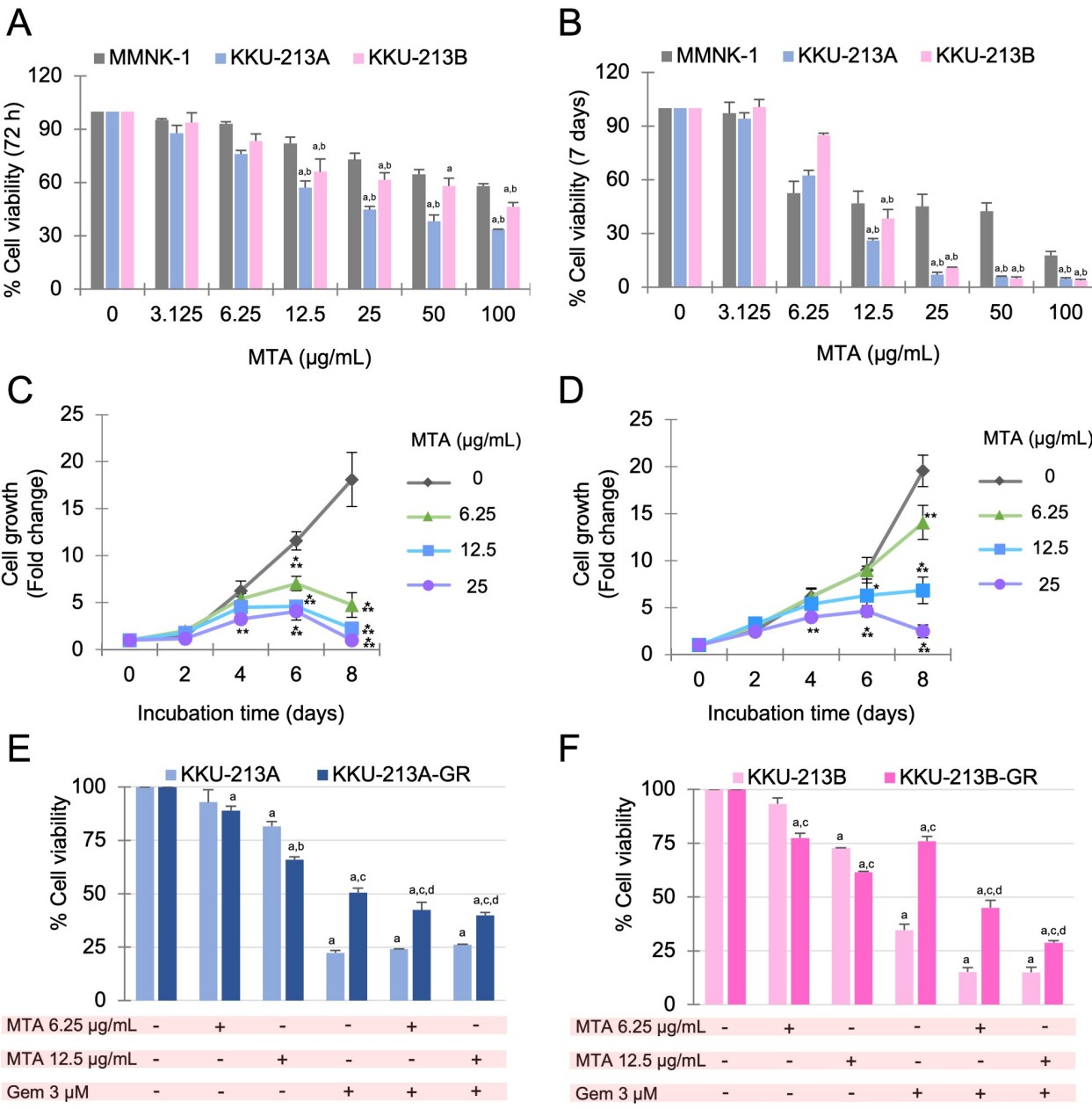

**Fig 2. Effect of MTA on cell viability and growth in CCA cell lines.** Cell cytotoxicity assay were performed on MMNK-1, KKU-213A and KKU-213B cells after treatment with various concentrations of MTA for 72 hours (A) and 7 days (B). Significance differences are indicated by 'a' for treatment vs. control and 'b' for CCA cell lines vs. the normal cell line. Cell growth assays were conducted on KKU-213A (C) and KKU-213B (D) cells treated with 0, 6.25, 12.5 and 25 μg/mL MTA, with MTT assays performed at day 0, 2, 4, 6, and 8. Statistical significance is denoted as $^*p<0.05$, $^{**}p<0.01$, $^{***}p<0.001$ when comparing MTA treatment vs. untreated control at each time point. The effects of MTA in combination with gemcitabine on parental cells and their drug-resistant counterparts were evaluated, with (E) showing the percentage of cell viability for KKU-213A and KKU-213A-GR treated with MTA alone and in combination with gemcitabine, and (F) displaying the percentage of cell viability for KKU-213B and KKU-213B-GR under the same conditions. The data are presented as the mean ± SD of three replicates. Significance differences are indicated by 'a' for treatment vs. control, 'b' for parental cells vs. drug-resistant cells, 'c' for MTA alone vs. MTA + gemcitabine, and 'd' for gemcitabine alone vs. MTA + gemcitabine.

**Table 2. The half-maximal inhibitory concentrations (IC$_{50}$) of MTA on MMNK-1, KKU-213A and KKU-213B at 72 hours and 7 days.**

| Cell line | IC$_{50}$ (µg/mL) of MTA | |
|---|---|---|
| | **72 hours** | **7 days** |
| **MMNK-1** | 234.70 ± 12.60 | 18.30 ± 4.80 |
| **KKU-213A** | 25.88 ± 3.07 | 8.79 ± 0.34 |
| **KKU-213B** | 77.56 ± 13.92 | 11.33 ± 0.50 |

results showed that MTA-treated cells had a higher percentage of apoptotic nuclei than the control in both KKU-213A (Fig 3B) and KKU-213B cells (Fig 3D).

## MTA suppresses migration and invasion capability of CCA cells

The effect of MTA on cell migration and invasion was investigated using concentrations of 12.5 and 25 µg/mL. The scratch assay was utilized to assess cell migration in two dimensions, while the modified Boyden chamber assay was used to determine both cell migration and invasion capability. The results from the scratch assay showed that the wound closure rate was significantly slower in MTA-treated cells in both KKU-213A (Fig 4A) and KKU-213B (Fig 4B).

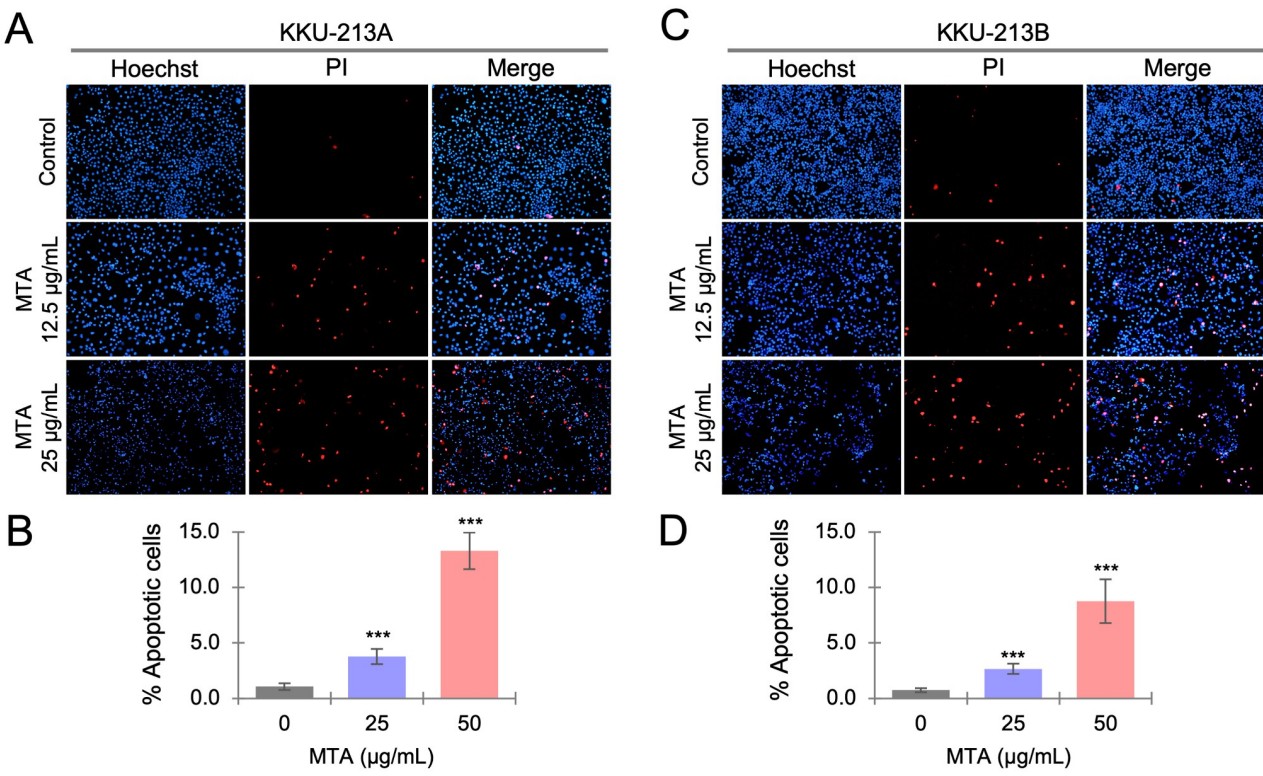

**Fig 3. Investigation of the induction effect of MTA on cell apoptosis.** The cell apoptosis-inducing effects of MTA were examined by using Hoechst/propidium iodide double staining to visualize apoptotic cells. KKU-213A and KKU-213B cell lines were treated with 12.5 and 25 µg/mL MTA for 72 hours. Representative images (A) show stained KKU-213A cells, and the chart (B) illustrates the percentage of apoptotic cells in each group. Representative images (C) depict stained KKU-213B cells, and the chart (D) presents the percentage of apoptotic cells in each group. The data represents means ± SD from six random low-power fields from triplicate wells. Significant differences between treated and untreated controls are indicated by *$p < 0.05$, **$p < 0.01$, and ***$p < 0.001$.

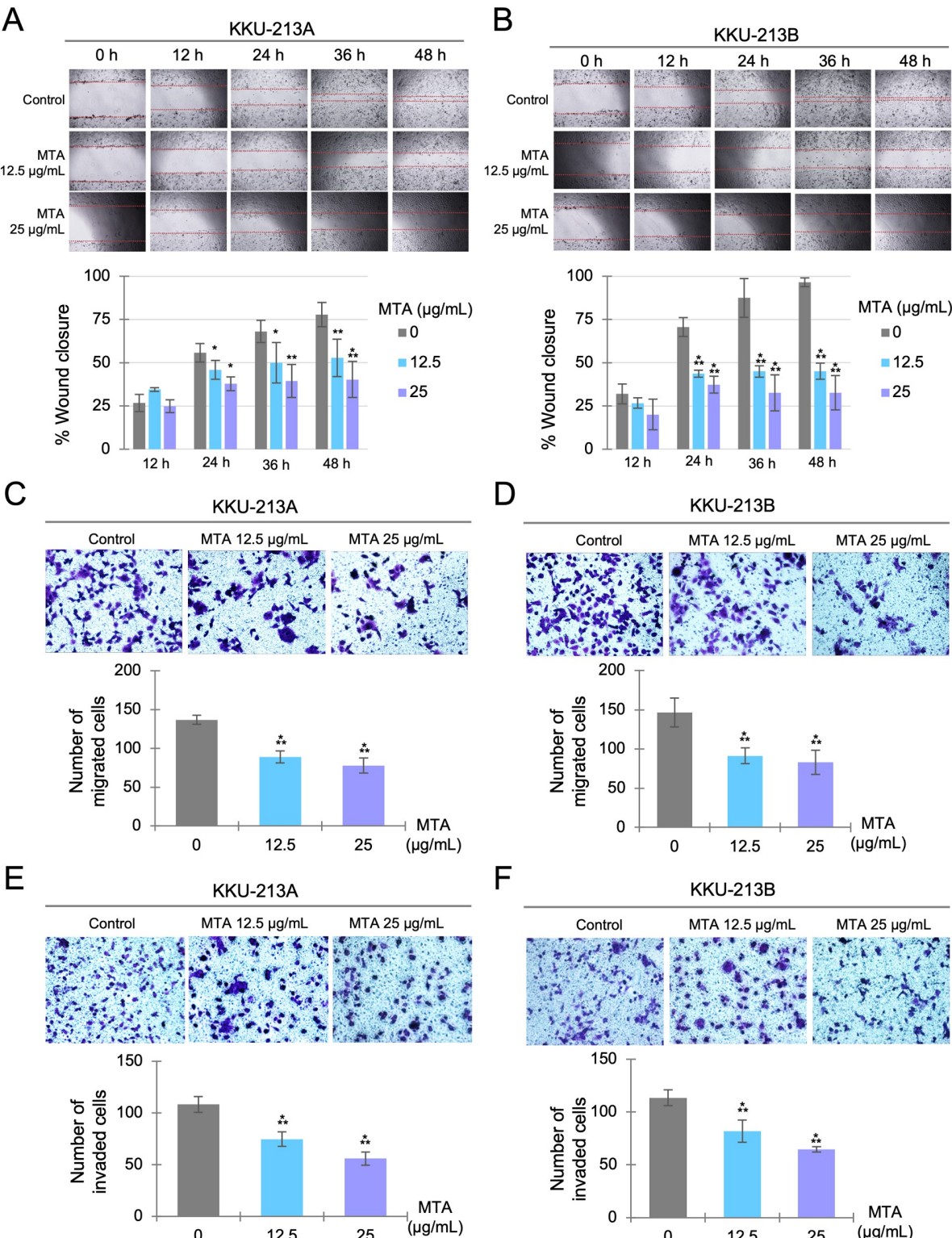

**Fig 4. Effects of MTA on cell migration and invasion.** Scratch assays were conducted with KKU-213A (A) and KKU-213B (B). Confluent cell monolayers were scratched and then treated with MTA, with representative images taken every 12 hours. The percentage of wound closure was calculated by comparing with 0 hour. The cell migration and invasion assays used a modified Boyden chamber, measuring cell migration in KKU-213A (C) and KKU-213B (D) and cell invasion in KKU-213A (E) and KKU-213B (F). The CCA cells were seeded and allowed to migrate or invade through the porous membrane of the Transwells® for 12 hours. The migrated or invaded cells were stained

and observed under a microscope, and the cells in six random low-power fields were counted and compared. The data are presented as mean ± SD from three replicates. Significant differences between treated and untreated controls are indicated by $^*p < 0.05$, $^{**}p < 0.01$, and $^{***}p < 0.001$.

Additionally, the modified Boyden chamber assay demonstrated that the number of migrated (Fig 4C and 4D) and invaded (Fig 4E and 4F) cells was significantly reduced in the MTA-treated groups for both KKU-213A and KKU-213B cells.

## Identification of differentially expressed proteins in MTA-treated cells using proteomic analysis

To further investigate the possible anticancer mechanisms of MTA, we conducted LC-MS/MS and bioinformatic analysis to identify differentially expressed proteins in MTA-treated samples. The summary workflow of the proteomic and data analysis is shown in Fig 5A. We identified unique, non-redundant proteins in both untreated control and MTA-treated cells across all biological triplicate samples. The results showed that 9 proteins were exclusively found in untreated control samples, 30 proteins were unique to MTA-treated KKU-213A samples, while 4,077 proteins were expressed in both treated and untreated samples (Fig 5B). Proteins with expression levels decreased or increased in MTA-treated cells by at least 2-fold and FDR < 0.05 were categorized as differentially expressed proteins (DEPs). Using these criteria, we identified 233 down-regulated and 38 upregulated proteins in MTA-treated KKU-213A cells when compared to the untreated controls (Fig 6A and S1 Table). In this study, we focused on the down-regulated protein populations because we aimed to identify the cancer-related pathways and potential target proteins that are inhibited by MTA treatment. Therefore, 233 down-regulated proteins were further analyzed.

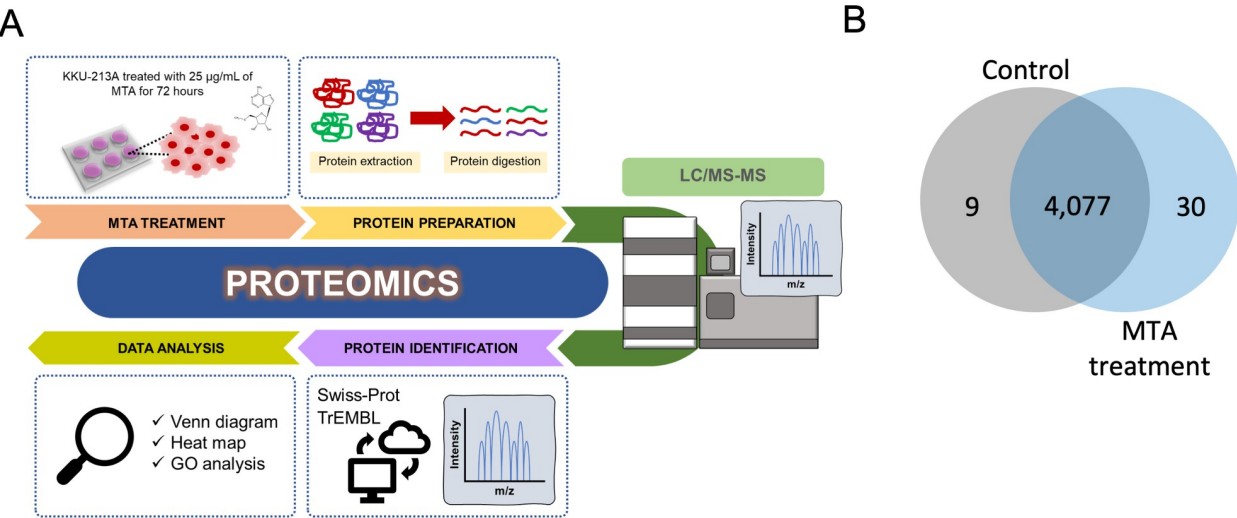

**Fig 5. Identification of differentially expressed proteins by proteomics analysis.** (A) An illustration of the proteomic experimental workflow. (B) A Venn diagram showing the proteins identified in KKU-213A control and MTA treatment.

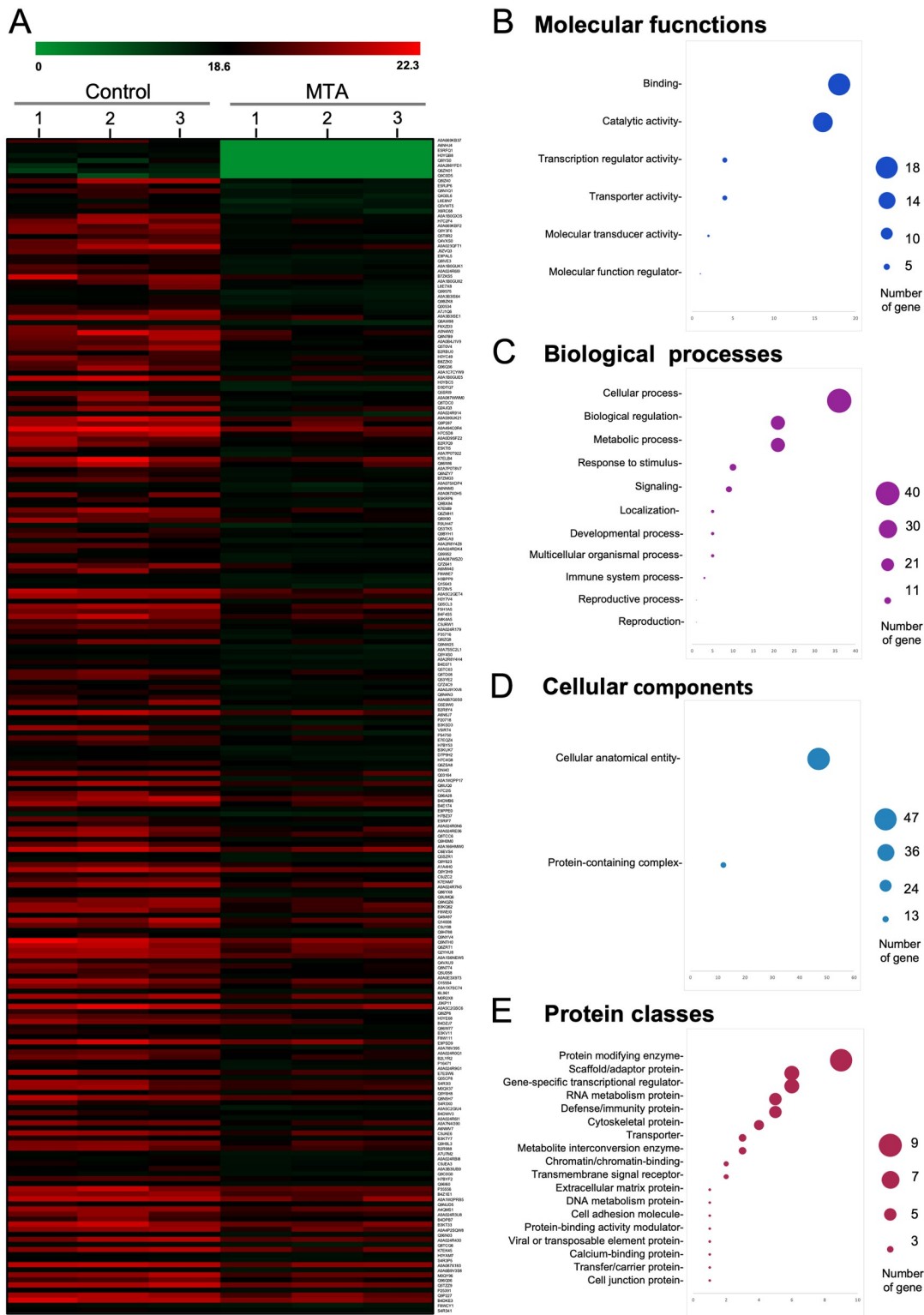

**Fig 6. Visualization and classification of downregulated protein after MTA treatment.** (A) Heatmap depicting proteins downregulated by at least 2-fold after MTA treatment. (B) Downregulated proteins after MTA treatment in KKU-213A categorized by Gene Ontology in terms of Molecular Function, (C) Biological Processes, (D) Cellular Component, and (E) Protein Class.

## Network analysis and functional enrichment analysis of down-regulated proteins

In our pursuit of a comprehensive understanding of proteomic changes induced by MTA treatment, we subjected the list of down-regulated proteins obtained from proteomic data to Gene Ontology (GO) analysis using the PANTHER (Protein Analysis Through Evolutionary Relationships) web portal. This allowed us to categorize the proteins based on their molecular function, biological process, cellular component, protein class, and pathway. The molecular function ontology highlighted primary protein functions, such as binding and transcription regulator activities, followed by additional functions, including transporter, molecular transducer, and molecular function regulator activity (Fig 6B). Among the down-regulated proteins, when categorized under biological processes, the largest fraction belonged to cellular processes, with other significant groups including biological regulation and metabolic processes (Fig 6C). Regarding cellular components, the down-regulated proteins were categorized into two main groups: cellular anatomical entities and protein-containing complexes (Fig 6D). The protein class ontology indicated that the majority of proteins belonged to the protein modifying enzyme class, followed by scaffold/adaptor proteins and gene-specific transcriptional regulator proteins (Fig 6E).

Next, we conducted further analysis of the down-regulated proteins to unveil protein networks. We identified four main interaction networks that visually represent the correlations between enriched pathways. Connections between two nodes were established when they share a minimum of 20% of gene numbers. Nodes with a darker shade indicate more significantly enriched gene sets, while larger nodes represent larger gene sets. Thicker lines signify a higher number of overlapping genes. In our network analysis of cellular components, we found that MTA treatment had a notable impact on proteins involved in the mitochondrial membrane (Fig 7A). Additionally, we identified three major networks related to cellular processes. These networks encompassed proteins associated with energy derivation, such as those involved in the electron transport chain, oxidative phosphorylation, ATP synthesis coupled electron transport, and the generation of precursor metabolites and energy, all of which were reduced after MTA treatment (Fig 7B). Furthermore, MTA treatment affected the protein trafficking network (Fig 7C), including processes such as vesicle-mediated transport, macromolecule localization, and lipid transport. Finally, proteins associated with changes in cell morphology were down-regulated, including those involved in cytoskeleton organization, cellular component morphogenesis, and cell projection organization (Fig 7D).

**mRNA expression of six candidate genes identified in proteomics study.** Based on the proteomics analysis, a set of candidate proteins were selected from the top 10 most down-regulated protein after MTA treatment. To validate these findings, we conducted RT-qPCR analysis and confirmed the decreased expression of 6 genes including ZNF860, KLC1, GRAMD1C, MAMSTR, TANC1, and TTC13. Remarkably, these genes displayed significantly reduced mRNA levels upon treatment with MTA (25 ug/mL) in both KKU-213A and KKU-213B cell lines (Fig 8A and 8B).

**KLC1 protein expression in MTA-treated cells and its mRNA expression in CCA patients' tissue.** Further analysis was conducted to validate the expression of KLC1 at the protein level in MTA-treated cells and its mRNA expression in patients' tissues based on TCGA data. Western blot results confirmed that KLC1 protein was indeed down-regulated in both CCA cell lines (KKU-213A and KKU-213B) following treatment with MTA compared to the untreated control (Fig 8C). Additionally, analyzing KLC1 mRNA expression in CCA patients' samples on the GEPIA2 web portal, the results showed that KLC1 was highly expressed in CCA tissues while expressed at a low level in normal bile duct tissues (Fig 8D).

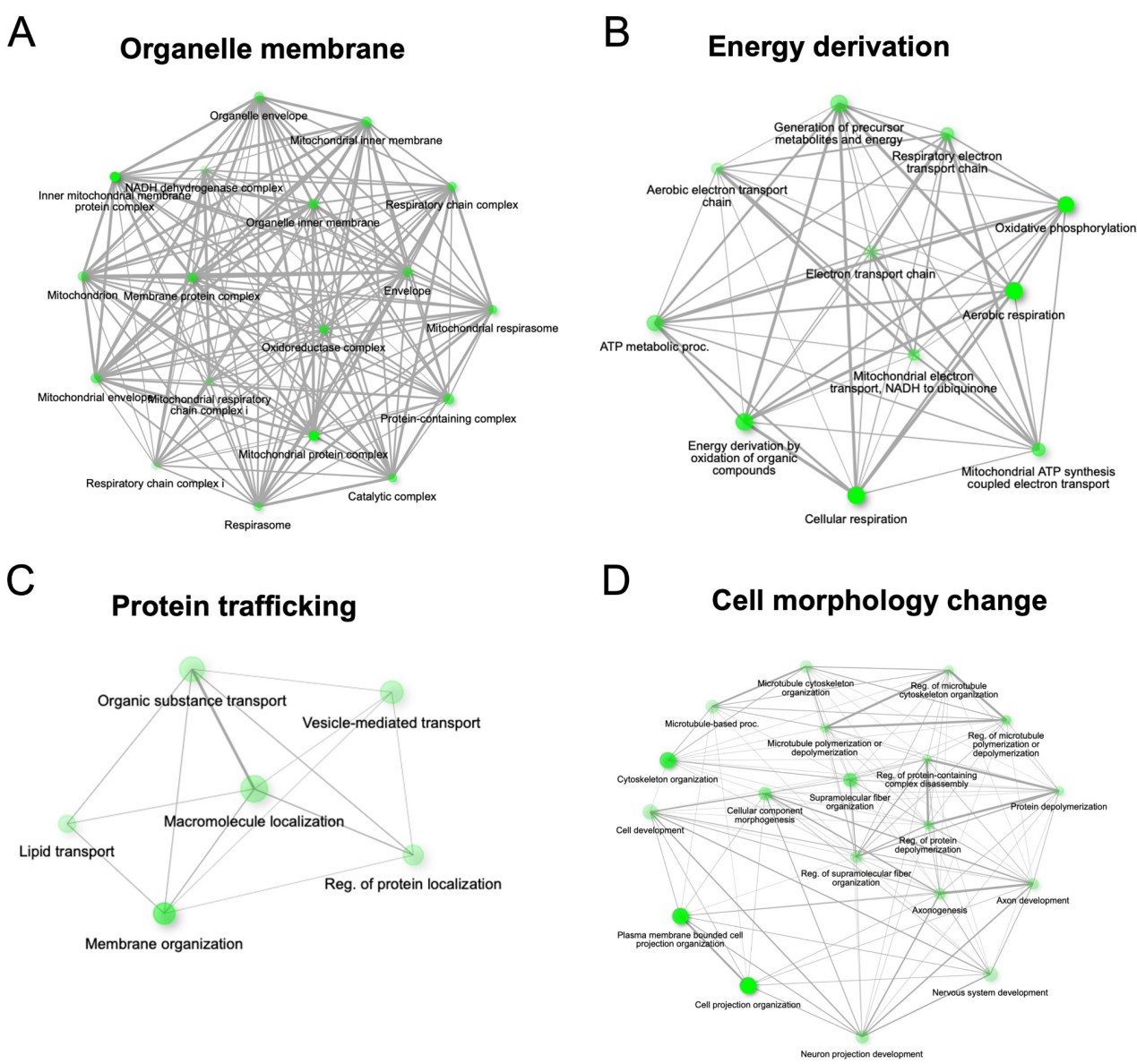

**Fig 7. Network analysis shows the relationship between enriched nodes.** Four protein networks of downregulated proteins were identified. The largest component of downregulated proteins was identified in the mitochondrial membrane (A). The major networks of cellular processes involved in energy derivation of the cell (B), protein trafficking (C), and cell morphology change (D).

## Discussion

In our current study, we assessed the effect of MTA on KKU-213A, KKU-213B, and their respective drug-resistant cell lines. Our data demonstrated that MTA treatment exerts a significant anticancer effect by inhibiting cell proliferation, migration, invasion, and inducing apoptosis in CCA cells. Notably, MTA treatment also augmented the anticancer efficacy of gemcitabine in drug-resistant cells. The results from proteomics analysis revealed that MTA effectively inhibited various proteins that have oncogenic properties, corroborating earlier findings.

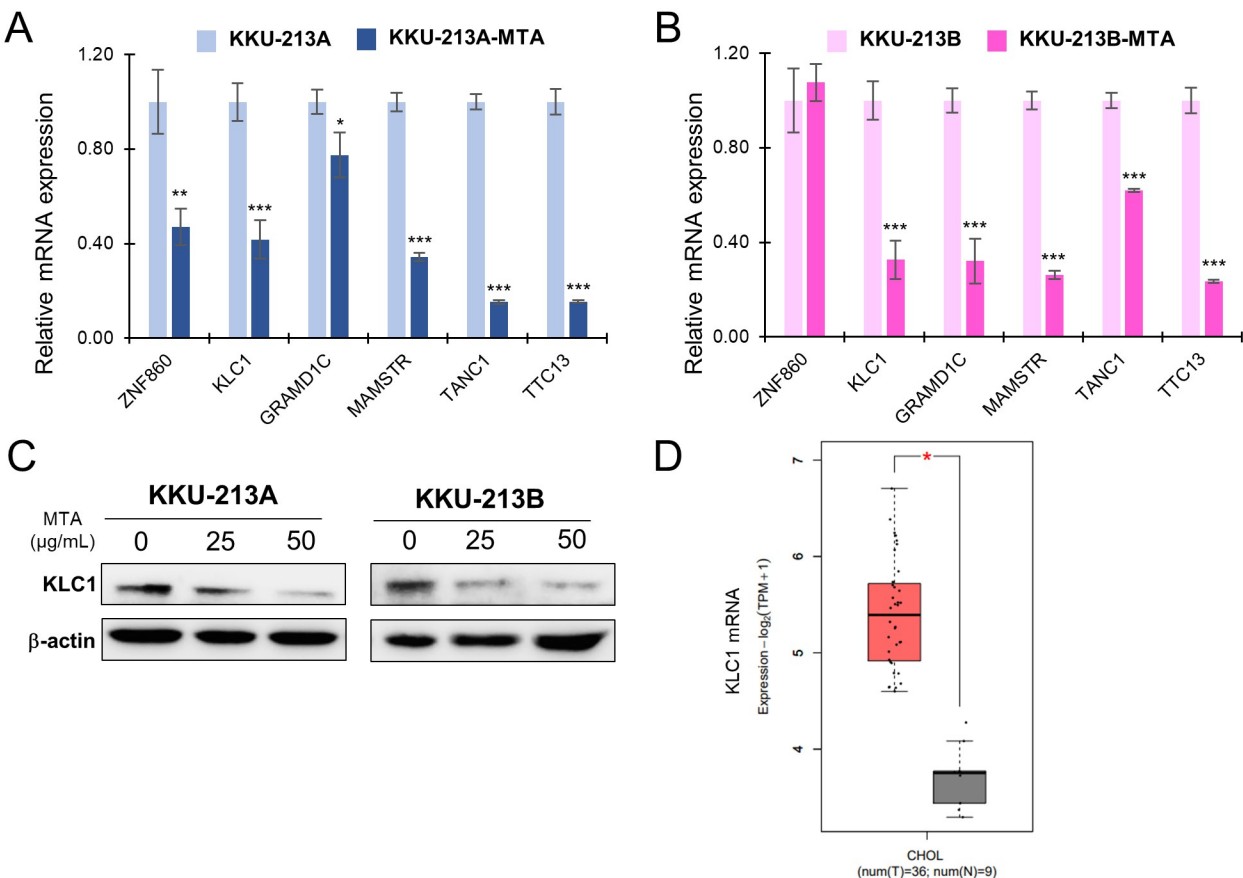

**Fig 8. Validation of candidate genes identified in proteomics analysis.** mRNA expression of six candidate genes was verified through RT-qPCR comparing between control and MTA-treated cells in KKU-213A (A) and KKU-213B (B). Significant differences between treated and untreated controls are indicated by $^*p< 0.05$, $^{**}p < 0.01$, and $^{***}p<0.001$. (C) KLC1 protein expression was verified in MTA-treated and untreated KKU-213A and KKU-213B cell lines. (D) KLC1 mRNA expression in CCA patients' samples comparing between tumor and normal adjacent tissues, analyzed via the GEPIA2 web portal based on TCGA data.

Gemcitabine is one of the first-line treatments for advanced CCA; however, clinical studies have shown that the overall survival and progression-free survival of patients receiving gemcitabine are still low [23]. Furthermore, a large number of patients experience relapse after receiving first-line chemotherapy, and the formidable side effects associated with these treatments contribute to suboptimal outcomes [24]. Consequently, we have been exploring naturally occurring agents with low toxicity toward normal cells, either for use as a single agent or combined with chemotherapeutic drugs to reduce the required chemotherapy dosage.

MTA, a compound known for some time, has recently garnered significant attention in the field of cancer research [25–27]. A number of studies have reported the effect of MTA on cancer cell viability, spanning various cancer types, such as melanoma, colorectal cancer, lymphoma, and erythroleukemia [11,28–31]. Notably, Tomasi and colleagues have identified MTA as a potential therapeutic agent against colorectal cancer, with minimal observed effects on normal cells [11]. Concomitantly, we found that MTA had a more profound effect on KKU-213A and KKU-213B in comparison to MMNK-1. The stronger impact on cancer cells may be attributed to the deficiency of 5-methylthioadenosine phosphorylase (MTAP), a key enzyme in the methionine salvage pathway (Fig 1B), which is also notably absent in most

cancers [10,12–14,32]. This deficiency leads to an abnormal adenine salvage pathway and the accumulation of unmetabolized MTA, resulting in cellular toxicity [9,33]. Interestingly, MTA also suppress cell viability in both gemcitabine resistant cell lines, KKU-213A-GR and KKU-213B-GR. The result show that when treated with MTA alone these cells were more sensitive to MTA than the parental cells (Fig 2E and 2F). Moreover, the combination of MTA and gemcitabine exhibited a more pronounced inhibitory effect on KKU-213A-GR and KKU-213B-GR than either agent alone (Fig 2E and 2F), suggesting that these two agents may have distinct downstream targets that complement each other in terms of cancer cell suppression. The previous research has studied the effect of MTA in combination with purine analogues such as 2'-fluoroadenine (2FA) in MTAP-deficient cancers. the result revealed that MTA combined with 2FA is more effective than 2FA alone [34]. In addition, another study showed that MTA used in combination with 5-FU enhances the anticancer activity of 5-FU [12]. This study is the first to present the impact of MTA on gemcitabine-resistant CCA cells when use singly or in combination with gemcitabine. Accordingly, MTA could potentially serve as a therapeutic solution against drug-resistant CCA in the future.

A primary goal of cancer treatment is to reduce the tumor burden and prevent the metastasis of cancer cells to distant organs, thereby prolonging patient survival. The treatment of late-stage tumors has traditionally relied on conventional chemotherapy and radiotherapy, which are known to be accompanied by significant and often debilitating side effects. Consequently, there has been a growing demand for novel therapeutic strategies capable of suppressing tumor metastasis with minimal side effects. Our study has revealed that MTA not only curtails the growth and induces apoptosis of CCA cells but, notably, it can also inhibit the invasion and migration of CCA cells at relatively lower concentrations. These findings are consistent with similar results observed in melanoma and hepatocellular carcinoma [29,35]. Furthermore, our proteomic analysis substantiates MTA's anticancer activity. Protein network analysis results suggest that MTA treatment leads to the reduction of proteins associated with changes in cell morphology, specifically those involved in cytoskeleton organization and the regulation of microtubule organization (Fig 7C). These morphological changes play a pivotal role in cancer cell migration and metastasis [36]. Collectively, our findings support the notion that MTA possesses the potential to suppress the invasion and migration capabilities of CCA cell lines.

Apoptosis, a self-regulated programmed cell death process, hinges on the activation, expression, and regulation of a multitude of proteins. Clinically, inducing cellular apoptosis represents a critical mechanism by which chemotherapeutic drugs control tumor progression [37]. Prior research has reported MTA's ability to induce apoptosis in various cancer cell lines, including HepG2, RKO, SW620, and U937 [11,38–40]. In our study, we further revealed that MTA could induce apoptosis of KKU-213A and KKU-213B in a dose-dependent fashion (Fig 3). Several pivotal events in the process of programmed cell death revolve around the mitochondria, encompassing the release of caspase activators [41], modifications in electron transport [42], loss of mitochondrial transmembrane potential [43], alterations in cellular oxidation-reduction processes [44], and the involvement of pro- and anti-apoptotic proteins [45]. Additionally, our network analysis, focusing on cellular components and biological processes, illustrates that down-regulated proteins are primarily implicated in mitochondrial function, encompassing the mitochondrial envelope, respiratory chain complex, inner mitochondrial membrane complex, respiratory electron transport chain, and also oxidative phosphorylation (Fig 7A and 7B).

Kinesin Light Chain 1 (KLC1) was initially identified as down-regulated in proteomics analysis and subsequently confirmed to be suppressed at both mRNA and protein levels following MTA treatments (Fig 8A–8C). Additionally, in the protein network analysis, KLC1 was

found to be involved in cell morphology changes (Fig 7D). Furthermore, it exhibited elevated expression levels in tumor tissues compared to normal tissues in CCA patients (Fig 8D). Conventionally, kinesin comprises two heavy chains and two light chains, forming a tetrameric structure. It facilitates the transport of various cargos towards the plus ends of microtubules, with the heavy chains providing motor activity and the light chains binding to different cargos, thus functioning as adapter molecules. KLC1 has been implicated in various cancer types by interacting with other proteins. For instance, ALK-KLC1 binding is associated with the metastasis and drug resistance of lung cancer [46–49]. In glioma, the fusion of KLC1 with ROS1 activates the JAK-STAT pathway [50,51]. KLC1 has also been found to interact with KIF5B, regulating cell-cell adhesion [52], and epithelial-mesenchymal plasticity in breast cancer [53]. Furthermore, KLC1 plays a role in the MAPK kinase pathway. However, the expression of KLC1 and its associated cargo in CCA has not yet been studied. This discovery suggests that MTA treatment suppresses KLC1 expression in CCA. Further detailed studies are encouraged to fully understand the implications of MTA and the role of KLC1 as a potential therapeutic target in CCA.

In conclusion, our findings strongly support the utility of MTA as an effective treatment option for advanced CCA. This is underscored by its ability to suppress the growth of CCA cells, curtail migration and invasion, and induce apoptosis, all while minimizing harm to normal cells. Additionally, MTA's synergistic impact with gemcitabine on drug-resistant cells further accentuates its therapeutic potential. Our proteomic analysis delves into the underlying mechanisms governing MTA's anticancer activity, with a particular emphasis on its ability to influence oncogenic proteins and modulate cellular morphology, and induce apoptosis through mitochondrial pathways. Collectively, our results endorse MTA as a promising strategy for advanced CCA treatment, effectively addressing concerns surrounding chemotherapy-induced side effects and potentially leading to improved patient outcomes.

## Supporting information

**S1 Table. Differentially expressed proteins in KKU-213A after MTA treatment.**
(DOCX)

**S1 Raw data. Raw data of cell viability, cell growth, drug-sensitivity assay, apoptosis assay, scratch assay, migration and invasion assays, RT-qPCR of MTAP and six candidate genes.**
(XLSX)

**S1 Raw images. Original blot images used for the generation of Fig 8C.** MS proteomics data. https://repository.jpostdb.org/entry/JPST002497; (PXID: PXD049119).
(PDF)

## Acknowledgments

The authors acknowledge the resources provided by the Central Equipment Center, Faculty of Science, Prince of Songkla University, specifically for the fluorescence microscope (cellSens Standard software, Olympus IX73 inverted microscope). The authors would also like to express gratitude to Miss Kawinnath Songsurin for her contribution to establishing the gemcitabine-resistant cell lines, KKU-213A-GR, and KKU-213B-GR.

## Author Contributions

**Conceptualization:** Sutthipong Nanarong, Thanawat Pitakpornpreecha, Phanthipha Runsaeng.

**Data curation:** Kankamol Kerdkumthong, Sutthipong Nanarong, Sittiruk Roytrakul.

**Formal analysis:** Kankamol Kerdkumthong, Sutthipong Nanarong, Sittiruk Roytrakul.

**Funding acquisition:** Sumalee Obchoei.

**Investigation:** Kankamol Kerdkumthong, Sutthipong Nanarong, Phonprapavee Tantimetta.

**Resources:** Sittiruk Roytrakul, Sumalee Obchoei.

**Supervision:** Sumalee Obchoei.

**Validation:** Sittiruk Roytrakul, Thanawat Pitakpornpreecha, Phanthipha Runsaeng, Sumalee Obchoei.

**Visualization:** Kankamol Kerdkumthong, Sutthipong Nanarong, Sumalee Obchoei.

**Writing – original draft:** Kankamol Kerdkumthong, Sutthipong Nanarong.

**Writing – review & editing:** Kankamol Kerdkumthong, Sutthipong Nanarong, Sittiruk Roytrakul, Thanawat Pitakpornpreecha, Phonprapavee Tantimetta, Phanthipha Runsaeng, Sumalee Obchoei.

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
