## [Decision Letter · Decision Letter 0]

27 Mar 2024

PONE-D-24-06933Quantitative proteomics analysis reveals possible anticancer mechanisms of 5'-deoxy-5'-methylthioadenosine in cholangiocarcinoma cellsPLOS ONE

Dear Dr. Obchoei,

Thank you for submitting your manuscript to PLOS ONE. After careful consideration, we feel that it has merit but does not fully meet PLOS ONE’s publication criteria as it currently stands. Therefore, we invite you to submit a revised version of the manuscript that fully addresses the points raised by the reviewer.

We look forward to receiving your revised manuscript.

Kind regards,

Matias A Avila, Ph.D.

Academic Editor

PLOS ONE

Journal Requirements:

"This research was supported by the National Science, Research and Innovation Fund (NSRF) and Prince of Songkla University (Ref. No. SCI6701024S) to Sumalee Obchoei. Kankamol Kerdkumthong was supported by a Prince of Songkla University-Ph.D. Scholarship (PSU_PHD2561-001). Phonprapavee Tantimetta was supported by a Prince of Songkla University-Ph.D. Scholarship (PSU_PHD2565-002)."

"The authors acknowledge the resources and support provided by the Central Equipment Center, Faculty of Science, Prince of Songkla University, specifically for the fluorescence microscope (cellSens Standard software, Olympus IX73 inverted microscope). The authors would also like to express gratitude to Miss Kawinnath Songsurin for her contribution to establishing the gemcitabine-resistant cell lines, KKU-213A-GR, and KKU-213B-GR."

Please remove any funding-related text from the manuscript. 

Reviewers' comments:

Reviewer's Responses to Questions

**Comments to the Author**

1. Is the manuscript technically sound, and do the data support the conclusions?

Reviewer #1: Partly

2. Has the statistical analysis been performed appropriately and rigorously? 

Reviewer #1: Yes

3. Have the authors made all data underlying the findings in their manuscript fully available?

Reviewer #1: Yes

4. Is the manuscript presented in an intelligible fashion and written in standard English?

Reviewer #1: Yes

5. Review Comments to the Author

Reviewer #1: I have carefully reviewed your manuscript on the anticancer activity of 5'-deoxy-5'-methylthioadenosine (MTA) in cholangiocarcinoma cell lines, focusing on the proteomic insights provided. Strengths of your paper include the thorough exploration of MTA's impact on cell viability, apoptosis, migration, and invasion in cholangiocarcinoma cell lines. To further enhance the robustness and impact of your study, I recommend considering the following points:

1.The expression of MTAP in MMNK and KKU-213A-B cell lines should be evaluated, as never has been published before. Regarding to this point, the reference you cite about the establishment of these cell lines is wrong (number #12). The correct one is the following: PMID: 32207095

2. You need to validate by western blot analysis the most representative genes associated with the pathways and processes identified in the proteomic study particularly concerning the biological effects of MTA on apoptosis, growth, viability, invasion, and migration of the cells you show. This validation approach will not only strengthen the findings of the study but also offer a deeper understanding of how these genes contribute to the cellular responses induced by MTA treatment.

3. Why did you choose KKU-213A for proteomic analysis? Which are the main differences between KKU-213A and KKU-213B cell lines?

4. The results in gemcitabine-resistant cell line experiments do not add any value to the work if they are not properly validated or discussed, so they need to be better discussed or otherwise removed from the manuscript.

Minor concerns:

1. Legend of figure 2: *p<0.05…etc statistical significances is it supposed to be between MTA and control-treated cells at every time-points? Please, write it properly.

2. The reference about gemcitabine resistant cells (Kerdkumthong et al, 2023) is not listed in the Bibliography section. Please, include it.

6. PLOS authors have the option to publish the peer review history of their article (what does this mean?). If published, this will include your full peer review and any attached files.

Reviewer #1: No

---

## [Author Response · Author response to Decision Letter 0]

16 May 2024

PONE-D-24-06933

Quantitative proteomics analysis reveals possible anticancer mechanisms of 5'-deoxy-5'-methylthioadenosine in cholangiocarcinoma cells

Journal Requirements:

1. Please ensure that your manuscript meets PLOS ONE's style requirements, including those for file naming. The PLOS ONE style templates can be found at…

Response: Thank you for providing us with the additional requirements for the revision of our manuscript. We have carefully reviewed your instructions and will ensure that all necessary adjustments are made to meet PLOS ONE's style and submission guidelines.

We will format our manuscript according to PLOS ONE's style requirements, including those for file naming. We have downloaded the PLOS ONE style templates from the provided links and will use them to format our manuscript appropriately.

"This research was supported by the National Science, Research and Innovation Fund (NSRF) and Prince of Songkla University (Ref. No. SCI6701024S) to Sumalee Obchoei. Kankamol Kerdkumthong was supported by a Prince of Songkla University-Ph.D. Scholarship (PSU_PHD2561-001). Phonprapavee Tantimetta was supported by a Prince of Songkla University-Ph.D. Scholarship (PSU_PHD2565-002)."

Response: Thank you for your suggestion. In this study, the funders had none of the mentioned roles, and we have stated this in the financial disclosure and included it in the cover letter. We thank you in advance for the changes you will make in the online submission form on our behalf.

"The authors acknowledge the resources and support provided by the Central Equipment Center, Faculty of Science, Prince of Songkla University, specifically for the fluorescence microscope (cellSens Standard software, Olympus IX73 inverted microscope). The authors would also like to express gratitude to Miss Kawinnath Songsurin for her contribution to establishing the gemcitabine-resistant cell lines, KKU-213A-GR, and KKU-213B-GR."

Please remove any funding-related text from the manuscript. 

Response: We have removed funding-related text from the Acknowledgments section of our manuscript as per your guidelines. We will only include acknowledgments for resources provided by the Central Equipment Center, Faculty of Science, Prince of Songkla University, and the contribution of Miss Kawinnath Songsurin in establishing the gemcitabine-resistant cell lines, KKU-213A-GR, and KKU-213B-GR.

Response: Captions for our Supporting Information files have been included at the end of our manuscript, and we have updated in-text citations to match accordingly. We adhered to PLOS ONE's Supporting Information guidelines to ensure proper formatting and presentation.

We appreciate your guidance in ensuring that our manuscript meets the journal's requirements. If you require any further clarification or have additional instructions, please do not hesitate to let us know.

Review Comments to the Author

Reviewer #1: I have carefully reviewed your manuscript on the anticancer activity of 5'-deoxy-5'-methylthioadenosine (MTA) in cholangiocarcinoma cell lines, focusing on the proteomic insights provided. Strengths of your paper include the thorough exploration of MTA's impact on cell viability, apoptosis, migration, and invasion in cholangiocarcinoma cell lines. To further enhance the robustness and impact of your study, I recommend considering the following points:

Response: We thank the reviewer for your thorough review of our manuscript. We truly appreciate your constructive feedback and have carefully considered each of your points.

1.The expression of MTAP in MMNK and KKU-213A-B cell lines should be evaluated, as never has been published before. Regarding to this point, the reference you cite about the establishment of these cell lines is wrong (number #12). The correct one is the following: PMID: 32207095

Response: We apologize for the incorrect reference citation. We have replaced the correct reference (PMID: 32207095) in our manuscript (ref no. 15, page 5, line 96 and bibliography no. 15, page 27, line 566-571). As per the reviewer’s advice, we acknowledge the importance of evaluating MTAP expression in these cell lines. We have measured the expression of MTAP in MMNK-1, KKU-213A, and KKU-213B cell lines as shown in Fig. 1B (page 4, line 85-86) and explained in the text in the Results section, pages 12-13, lines 253-257. The MTAP primer and internal control gene primer sequences are listed in table 1 (page 11, line 227). The RT-qPCR protocol was added in Material and method under the ‘RT-qPCR’ subsection (page 10-11, lines 216-225).

2. You need to validate by western blot analysis the most representative genes associated with the pathways and processes identified in the proteomic study particularly concerning the biological effects of MTA on apoptosis, growth, viability, invasion, and migration of the cells you show. This validation approach will not only strengthen the findings of the study but also offer a deeper understanding of how these genes contribute to the cellular responses induced by MTA treatment.

Response: We agree with the reviewer that validating proteomic findings is essential for strengthening the robustness of our study. Firstly, we selected a set of genes (6 genes) from the top 10 most down-regulated proteins from proteomics results, then verified them by RT-qPCR. The results are shown in Figs 8A and 8B (page 20, lines 423-425). In addition, we selected KLC1 for further verification via Western blot analysis for three reasons: 1. KLC1 mRNA was strongly down-regulated in both KKU-213A and KKU-213B after MTA treatment, 2. KLC1 mRNA was upregulated in CCA patients’ tissues compared with normal adjacent tissues based on TCGA data, suggesting its possible significant role in CCA, and 3. The verified antibody against KLC1 is available in the market. The Western blot results of the protein expression of KLC1 in MTA treated cells compared to the control are shown in Fig. 8C (page 20, lines 426-427). The protocol and antibodies used were stated in Material and method under the ‘Western blotting’ subsection (page 11-12, lines 229-240).

3. Why did you choose KKU-213A for proteomic analysis? Which are the main differences between KKU-213A and KKU-213B cell lines?

Response: Thank you for the question. KKU-213A and KKU-213B are both established from the tissue of the same patient (a Thai patient with intrahepatic cholangiocarcinoma), and these cell lines are widely used in the CCA research community. The tumor of origin of these cell lines was adenosquamous carcinoma. The major differences between these cell lines are distinct cell morphology, cytogenetic characteristics, and progressive phenotypes. In addition, when injected into the mice, KKU-213A formed poorly differentiated squamous cell carcinomas, while KKU-213B formed well-differentiated squamous cell carcinomas. We chose KKU-213A for proteomics analysis because the observed MTA effect on cell growth and apoptosis was stronger in KKU-213A compared to KKU-213B. However, the later verification of candidate gene expression was done in both cell lines.

4. The results in gemcitabine-resistant cell line experiments do not add any value to the work if they are not properly validated or discussed, so they need to be better discussed or otherwise removed from the manuscript.

Response: We thank the reviewer for the valuable comment. We understand your concern regarding the gemcitabine-resistant cell line experiments and agree that they need to be better validated or discussed to add value to the manuscript. The reason we investigated the effect of MTA on gemcitabine-resistant CCA cell lines was because drug resistance is one of the major challenges for CCA treatment. We also would like to emphasize that, in the future, MTA may be developed as a treatment option to combat drug-resistant CCA. As per the reviewer's advice, we have discussed this matter in the Discussion section (page 22, lines 455-467).

Minor concerns:

1. Legend of figure 2: *p<0.05…etc statistical significances is it supposed to be between MTA and control-treated cells at every time-points? Please, write it properly.

2. The reference about gemcitabine resistant cells (Kerdkumthong et al, 2023) is not listed in the Bibliography section. Please, include it.

Response: 1. Thank you for your comment. As per the reviewer's suggestion, we have fixed the legend of Figure 2 for clarification (page 14, lines 267-287).

Response: 2. Thank you for your comment. We apologize for the oversight, and we have included the reference (Kerdkumthong et al, 2023) in the Bibliography section of our manuscript (ref no. 17, page 5, line 100 and bibliography no. 17, page 27, lines 572-575).

Once again, we sincerely appreciate your valuable feedback, which will undoubtedly improve the quality and impact of our study.

---

## [Decision Letter · Decision Letter 1]

11 Jun 2024

Quantitative proteomics analysis reveals possible anticancer mechanisms of 5'-deoxy-5'-methylthioadenosine in cholangiocarcinoma cells

PONE-D-24-06933R1

Dear Dr. Obchoei,

We’re pleased to inform you that your manuscript has been judged scientifically suitable for publication and will be formally accepted for publication once it meets all outstanding technical requirements.

Kind regards,

Matias A Avila, Ph.D.

Academic Editor

PLOS ONE

Additional Editor Comments (optional):

Reviewers' comments:

Reviewer's Responses to Questions

**Comments to the Author**

1. If the authors have adequately addressed your comments raised in a previous round of review and you feel that this manuscript is now acceptable for publication, you may indicate that here to bypass the “Comments to the Author” section, enter your conflict of interest statement in the “Confidential to Editor” section, and submit your "Accept" recommendation.

Reviewer #1: All comments have been addressed

2. Is the manuscript technically sound, and do the data support the conclusions?

Reviewer #1: Yes

3. Has the statistical analysis been performed appropriately and rigorously? 

Reviewer #1: Yes

4. Have the authors made all data underlying the findings in their manuscript fully available?

Reviewer #1: Yes

5. Is the manuscript presented in an intelligible fashion and written in standard English?

Reviewer #1: Yes

6. Review Comments to the Author

Reviewer #1: I am pleased to inform you that your manuscript has met all the requirements and addressed the reviewers' comments satisfactorily.

7. PLOS authors have the option to publish the peer review history of their article (what does this mean?). If published, this will include your full peer review and any attached files.

Reviewer #1: No

---

## [Editor Report · Acceptance letter]

18 Jun 2024

PONE-D-24-06933R1 

PLOS ONE

Dear Dr. Obchoei, 

I'm pleased to inform you that your manuscript has been deemed suitable for publication in PLOS ONE. Congratulations! Your manuscript is now being handed over to our production team.

Kind regards, 

on behalf of

Dr Matias A Avila 

Academic Editor

PLOS ONE